# Validation of the STANDARD Q COVID-19 antigen test in Vojvodina, Serbia

**Mioljub Ristić**[1,2]*, **Nataša Nikolić**[2,3], **Velibor Čabarkapa**[4,5], **Vesna Turkulov**[6,7], **Vladimir Petrović**[1,2]

**1** Department of Epidemiology, University of Novi Sad, Faculty of Medicine, Novi Sad, Serbia, **2** Institute of Public Health of Vojvodina, Novi Sad, Serbia, **3** Department of Microbiology with Parasitology and Immunology, University of Novi Sad, Faculty of Medicine, Novi Sad, Serbia, **4** Faculty of Medicine, Department of Pathophysiology and Laboratory Medicine, University of Novi Sad, University of Novi Sad, Novi Sad, Serbia, **5** Centre of Laboratory Medicine, Clinical Centre of Vojvodina, Novi Sad, Serbia, **6** Department of Infectious Diseases, Faculty of Medicine, University of Novi Sad, Novi Sad, Serbia, **7** Clinic for Infectious Diseases, Clinical Centre of Vojvodina, Novi Sad, Serbia

* mioljub.ristic@mf.uns.ac.rs

**Data Availability Statement:** All relevant data are within the manuscript.

**Funding:** This work is a part of the research that was supported by Provincial Secretariat for Higher Education and Scientific Research grant number

## Abstract

### Background

Since COVID-19 pandemic is a global crisis, tests with high sensitivity and specificity are crucial for the identification and management of COVID-19 patients. There is an urgent need for low-cost rapid antigen COVID-19 test with a good diagnostic performance. Although various antigen rapid detection tests are widely available, strong evidence of their usefulness in clinical practice are still limited. Therefore, our aim was to evaluate clinical performance of STANDARD Q COVID-19 Ag Test (SD Biosensor, Gyeonggi-do, South Korea).

### Methods

The performance of the STANDARD Q COVID-19 Ag Test for the detection of SARS-CoV-2 antigen was evaluated in comparison to RT-qPCR results in 120 symptomatic patients (median age 49, IQR 36–70) who presented to health care facility in Novi Sad, Vojvodina, Serbia.

### Results

Twenty five out of 120 samples have been tested positive using STANDARD Q COVID-19 Ag Test, and all of them were also positive on RT-qPCR. Overall, the STANDARD Q COVID-19 Ag Test showed sensitivity of 58.1% (95% CI 42.1–73.0) but it was higher in the early days of disease, when the highest viral loads were detected. During the first five days after the symptom onset, the sensitivity ranged from 66.7% to 100% and the pooled accuracy and Kappa values were high (0.92 and 0.852).

### Conclusions

A strong agreement between performance of STANDARD Q COVID-19 Ag Test and RT-qPCR was observed during the first five days of illness, suggesting that this rapid antigenic test can be very useful for COVID-19 diagnosis in the early phase of disease.

142-451-3072/2020-03. The funders had no role in study design, data collection and analysis, decision to publish, or preparation of the manuscript.

## Introduction

On January 20th, China's "National Infectious Diseases Law" was amended to make 2019-novel coronavirus diseases (COVID-19) a Class B notifiable disease and its "Frontier Health and Quarantine Law" was revised to support the COVID-19 outbreak response effort [1]. Shortly thereafter, the COVID-19 pandemic, caused by severe acute respiratory syndrome coronavirus 2 (SARS-CoV-2), has become an ongoing global health crisis. The severity of COVID-19 symptoms range from very mild to severe pneumonia with multi-organ dysfunction and death [2, 3]. Due to the rapid spread of the SARS-CoV-2, on January 30th, WHO declared COVID-19 outbreak a public health emergency of international concern [3, 4]. As of November 4th, 2020, a total of 47,582,064 cases were confirmed worldwide causing a total 1,217,540 deaths [5].

Diagnostic tests of high sensitivity and specificity are crucial for the identification and management of COVID-19 patients. In particular, high diagnostic accuracy tests applied in the early phase of the illness would enable identification of COVID-19 patients and promptly implementation of control measures in order to reduce household and community transmission [2, 6, 7]. To this end, "point-of-care" or "near patient" antigen and molecular tests for detection of a current SARS-CoV-2 infection have the potential to allow fast laboratory confirmation and timely isolation of COVID-19 cases [6].

The gold standard for laboratory confirmation of SARS-CoV-2 infection is the quantitative real-time reverse transcription polymerase chain reaction test (RT-qPCR) [8, 9], but this technique is expensive and usually takes at least 24 hours to produce the result, making its implementation challenging in many countries [6]. On the other hand, reliable but less expensive and faster diagnostic tests have been developed to specifically detect antigens of SARS-CoV-2 virus. These rapid diagnostic tests, or RDTs, are designed to directly detect SARS-CoV-2 proteins produced by replicating virus in respiratory secretions, and they are developed for both laboratory-based and "near patient"("while you wait") use [6, 10]. So far, strong evidence of the usefulness of these tests in clinical practice are still largely lacking [6].

Therefore, the aim of the present study was to evaluate the performance of a STANDARD Q COVID-19 Ag TestSD Biosensor, Gyeonggi-do, South Korea among symptomatic patients in the early and late phases of the disease [6, 11].

The findings of this study provide an insight into the usefulness of this test, with special attention given to the potential (at least partial) replacement of the RT-qPCR test for confirmation of current SARS-CoV-2 infection in the first days of infection.

## Materials and methods

### Study design

Analysis of prospectively collected data, at the triage ambulance of primary and tertiary outpatients health care facility in Novi Sad (the capital of the autonomous province of Vojvodina, Serbia), was conducted. The outpatients were enrolled by Health Centre Novi Sad (COVID ambulance of primary health care level) and by the ambulance ("red zone") of Clinical Centre of Vojvodina, Department for Infectious Diseases, from 21st August to 1st September, 2020. The education about adequate sampling and the samples handling procedures of all included physicians and nurses was conducted before the start of research and lasted for 10 days.

**Inclusion/Exclusion criteria.** We included all symptomatic patients suspected of SARS-CoV-2 infections presenting with one or more of the following signs/symptoms: fever, malaise, cough, sore throat, myalgia, arthralgia, headache, coryza, diarrhoea, nausea, anosmia or ageusia, vomiting), regardless of their age and gender. Asymptomatic patients were excluded from

the analysis. Physicians interviewed patients (who met the inclusion criteria) through face -to-face structured interviews on the day of the sampling. The questionnaire was comprised of questions about sociodemographic factors, date of the symptoms onset and the severity (mild or moderate) of clinical signs and symptoms related to SARS-CoV-2 infection.

**Specimen collection and laboratory testing.** Posterior nasopharyngeal (PNP) swabs were collected. Using STANDARD Q COVID-19 Ag Test, PNP samples were tested immediately after collection, on-site using STANDARD Q COVID-19 Ag Test, by previously trained medical staff, as previously described in study design section. The transport of clinical samples in a hand refrigerator (at +2- +8˚C) from the spot of sampling to the laboratory of the Centre for Virusology of Institute of Public Health of Vojvodina, Novi Sad that performed analyses was organized on a daily basis. The samples were accompanied with the previously completed questionnaires.

For RT-qPCR laboratory confirmation, the PNP swabs were transported in sterile commercially available tubes containing a specific viral transport medium with antifungal and antibiotic supplements. Samples were, held refrigerated at 4˚C and tested within 12 hours of collection. Before testing, in order to reduce the risk of accidental transmission of SARS-CoV-2 to laboratory staff, the PNP swabs were inactivated by heat in a water-bath, at 56˚C for 35 ± 5 minutes [9].

*Molecular detection by RT-PCR*. Each sample was initially examined for detection of SARS-CoV-2 genes by Argene®, SARS-COV-2 R-GENE® assay (bioMérieux, Marcy-l'Ètoile, France), after RNA extraction on QIAcube automated workstation (Qiagen, Hilden, Germany). Automated extraction of nucleic acid was done with QIAamp Viral RNA Mini Kit in adapters of up to 12 samples. A two-step approach was used, a qualitative RT-PCR followed by the quantitative one focused on viral loads. Firstly, samples were screened for SARS-CoV-2 RNA by qualitative RT-PCR, targeting three regions that had conserved sequences: the RdRP gene (RNA-dependent RNA polymerase gene) in the open reading frame ORF1ab region, the E gene (envelope protein gene), and the N gene (nucleocapsid protein gene). Reaction, amplification conditions, and results interpretation were performed according to the manufacturer's instructions. A qualitative real time assay was performed on the Applied Biosystems 7500 Real-Time PCR System (Life Technologies, Carlsbad, CA, USA) with software version 2.3. Samples showing an exponential growth curve with any cycle threshold (Ct) value were considered positive.

Secondly, in order to detect viral load, SARS-CoV-2 positive samples were additionally analyzed by quantitative RT-PCR (RT-qPCR). COVID-19 Genesig Real-Time PCR Kit (Primerdesign Ltd, Chandler's Ford, UK) was utilized for RT-qPCR assay. The experiment was designed according to the manufacturer's instructions. The target viral gene is the RdRP and the limit of detection (LoD) reported by the manufacturer is 0.58 copies/μL. The reverse transcription and amplification were performed on the Applied Biosystems 7500 Real-Time PCR System with software version 2.3. Quantification of the number of RNA copies was done according to a scale ranging from 2 to $2 \times 10^5$ copies per μL of positive control used as the manufacturer's standards. The cycle threshold value (Ct) of the positive sample is compared with the standard curve to determine the viral load (viral copy number) in the sample, which was expressed as copy number per μL or mL. The PNP swabs with SARS-CoV-2 RT-qPCR Ct under 41 were considered positive.

**STANDARD Q COVID-19 Ag Test.** STANDARD Q COVID-19 Ag Test (SD Biosensor, Gyeonggi-do, South Korea) is a chromatographic immunoassay. It allows rapid (in 15–30 minutes) and qualitative detection of SARS-CoV-2 proteins in nasopharyngeal swabs. This test is intended to be used in patients with clinical symptoms of SARS-CoV-2 infection. Test has two pre-coated lines, namely "C" line (Control line), and "T" line (Test line) on the surface of the

nitrocellulose membrane. No lines in the result window are visible prior to applying any speci-
mens. This STANDARD Q COVID-19 Ag Test consists of the mouse monoclonal anti-SARS-
CoV-2 antibody (coated in the test line region) and the mouse monoclonal anti-Chicken IgY
antibody (coated in the control line region). During the test, SARS-CoV-2 antigen in the speci-
men interacts with monoclonal anti-SARS-CoV-2 antibody conjugated with colour particles,
resulting in a visible coloured antigen-antibody complex, without possibility for interaction
with any disruptive agents. This complex migrates to the membrane via capillary action
towards the test line, where it is captured by the mouse monoclonal anti-SARS-CoV-2 anti-
body. Consequently, a coloured test line appears in the result window if SARS-CoV-2 antigens
are present in the specimen. The intensity of the coloured test line varies depending on the
amount of SARS-CoV-2 antigen present in the specimen. If SARS-CoV-2 antigens are not
present in the specimen, then no colour will appear in the test line. The control line is used as a
quality control and should always appear if the test procedure is performed properly and the
test reagents are valid [11].

## Data analysis

Parametric and non-parametric, correlative, linear and nonlinear regressive analyses were per-
formed using SPSS software tool (version 22) MedCalc for Windows, version 12.3.0 (MedCalc-
Software, Mariakerke, Belgium). Different statistical analyses were performed for different
variables, according to the type of variable: Kruskal-Wallis H test for continuous, non-
parametric (ordinal scale) variables, and Fisher's exacttest or chi-square for categorical data.
To determine the predictive validity of the STANDARD Q COVID-19 Ag Test and the level of
its agreement with the RT-qPCR test, sensitivity (Se), specificity (Sp), positive predictive value
(PPV), negative predictive value (NPV), accuracy, Kappa coefficient (Kappa) and their 95% CI
were calculated. Results of the STANDARD Q COVID-19 Ag Test were compared to those of
RT-qPCR, which was considered the gold standard for this evaluation (positive and negative
results obtained by RT-qPCR were considered to be true positive and true negative results,
respectively). Test with an accuracy value above 0.71 was deemed useful, and above 0.90 was
considered as being of high diagnostic value [12, 13]. Value of Kappa of $\geq 0.600$ was defined
as moderate level of agreement between the two tests [14]. Statistical significance was set at the
value of $p < 0.05$.

## Ethical considerations

This quality assurance study was conducted as a part of the daily clinical routine practice. Oral
informed consent for the research was obtained from all patients at the moment of sampling
that took place in the period from August, 21st to September, 1st, 2020, in accordance with
national regulations. The ethical agreement for this research was obtained from the Ethics
Committee of the Institute of Public Health of Vojvodina, Novi Sad and from the Ethical Com-
mittee of the Faculty of Medicine, University of Novi Sad. No authors of this study were
involved in the treatment of the patients included in the analysis, and all data were anon-
ymized before the authors accessed it.

## Results

A total of 120 participants were included in the study. The prevalence of various signs and
symptoms as well as the details about the time elapsed between disease onset and sample col-
lection, by age of the participants are shown in Figs 1 and 2, respectively. The most common
(103 of 120, 85.8%) sign/symptom among all participants was fever (not measured at the spot),
while the least frequent one (8 of 120, 6.7%) was vomiting.

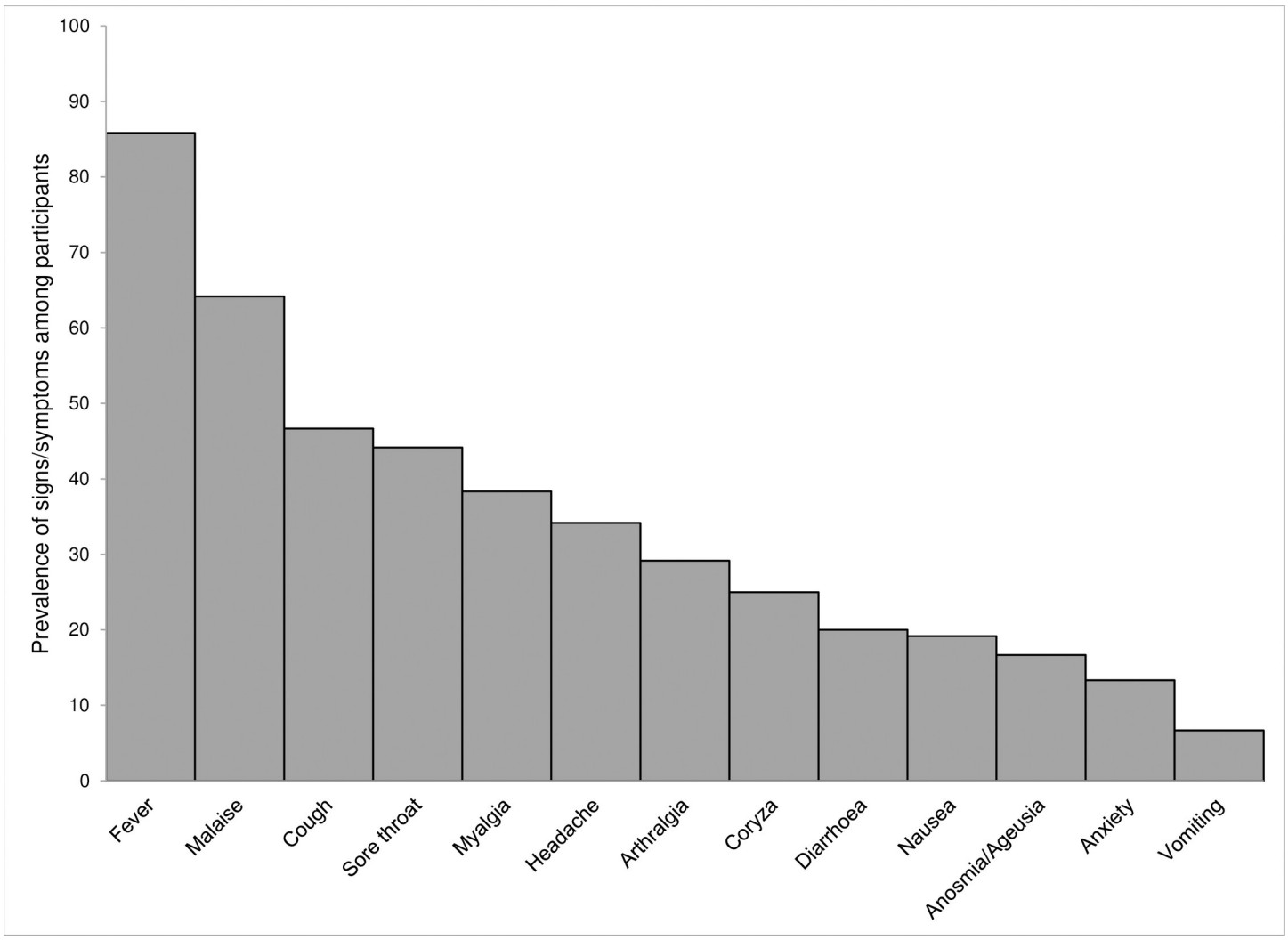

**Fig 1. Prevalence of signs and symptoms among participants tested for COVID-19.**

Median age of observed patients was 49 years (IQR 36–70), with the youngest participant being 14, and the oldest one 91 years of age. The average period between signs/symptoms onset and swab collection was 9.4 days (ranging between 1 and 45 days) and the median time was 5 days (IQR 3–15). Out of all participants, 52.5% (63/120) cases were tested within the first five days after symptoms onset.

Twenty five out of 120 samples have been tested positive using STANDARD Q COVID-19 Ag Test, and all of them were also positive on RT-qPCR. On the other hand, 35.8% (43/120) were tested positive by RT-qPCR for SARS-CoV-2 infection. Furthermore, we assessed association of RT-qPCR confirmation of SARS-CoV-2 with age, days elapsed from symptom onset, results of rapid test results and the frequency of various signs/symptoms (Table 1). As a result, RT-qPCR confirmation of SARS-CoV-2 in tested respondents was statistically significant (p<0.05) associated with middle age, use of the STANDARD Q COVID-19 Ag Test and certain (cough, malaise, anosmia/ageusia, myalgia, arthralgia and headache) signs/symptoms.

We also stratified Se, Sp, PPV, NPV, accuracy and Kappa of the STANDARD Q COVID-19 Ag Test with regard to certain characteristics based on comparison with the laboratory

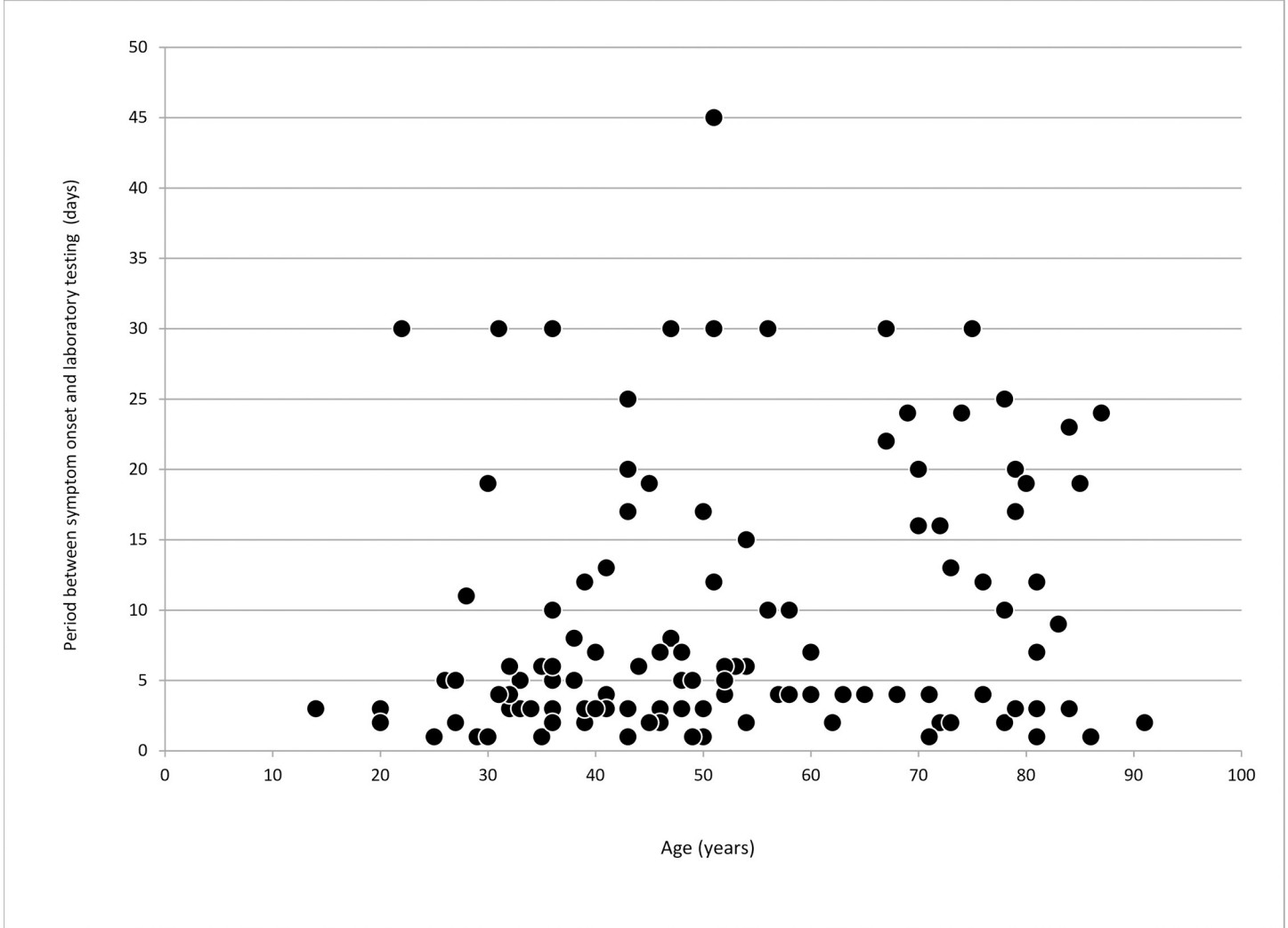

**Fig 2. Scatter plot of period between signs and symptoms onset and swab collection for laboratory testing for COVID-19 by age.**

confirmation of SARS-CoV-2 infection by RT-qPCR. The optimum values of 100% of Sp and PPV were observed regarding gender, age group, and days of signs/symptoms onset. Values of accuracy and Kappa were slightly higher in females (0.89 and 0.689) than in males (0.81 and 0.597), as well as in the participants aged 14–30 years (0.92 and 0.755), compared to the other two age groups, namely 31–64 year group (0.85 and 0.677) and the ≥65 year group (0.83 and 0.341). Concerning the time period between the onset of disease and the sampling, we registered the maximum values of observed performance for STANDARD Q COVID-19 Ag Test between the first and fifth day after symptom onset. Although the overall Se for, STANDARD Q COVID-19 Ag Test was 58.1% (95% CI 42.1–73.0), the values of Se were higher during the first five days following signs/symptoms onset (100%, 83.3%, 66.7%, 71.4% and 100%, respectively) than in later days (55.6% for period between 6th and 10th day, 50% for 11-15th day period, and 0% for more than 16 days after signs/symptoms onset). In the line with this, the pooled accuracy and Kappa values were higher in the first five days of disease (0.92 and 0.852, respectively) compared to the overall observed period-from 1 to 45 days between signs/symptoms onset and swab collection (0.85 and 0.641, respectively) (Table 2).

**Table 1. Demographic characteristics, STANDARD Q COVID-19 Ag Test and signs/symptoms associated with RT-qPCR laboratory confirmed SARS-CoV-2 infections.**

| Characteristics | | Overall (n = 120) | | RT-qPCR positive (n = 43) | | RT-qPCR negative (n = 77) | | p value [a] |
|---|---|---|---|---|---|---|---|---|
| | | No | % | No | % | No | % | |
| **Gender** | Male | 63 | 52.5 | 28 | 65.1 | 35 | 45.5 | 0.056 |
| | Female | 57 | 47.5 | 15 | 34.9 | 42 | 54.5 | |
| **Age group** | 14–30 | 13 | 10.8 | 3 | 7.0 | 10 | 13.0 | **0.040** [b] |
| | 31–64 | 71 | 59.2 | 32 | 74.4 | 39 | 50.6 | |
| | ≥65 | 36 | 30.0 | 8 | 18.6 | 28 | 36.4 | |
| **Days from symptom onset** | 1–5 | 63 | 52.5 | 24 | 55.8 | 39 | 50.6 | 0.544 [b] |
| | 6–10 | 20 | 16.7 | 9 | 20.9 | 11 | 14.3 | |
| | 11–15 | 8 | 6.7 | 2 | 4.7 | 6 | 7.8 | |
| | ≥16 | 29 | 24.2 | 8 | 18.6 | 21 | 27.3 | |
| **Results of STANDARD Q COVID-19 Ag Test** | Positive | 25 | 20.8 | 25 | 58.1 | 0 | 0.0 | **<0.001** [b] |
| | Negative | 95 | 79.2 | 18 | 41.9 | 77 | 100.0 | |
| **Signs/symptoms** | Fever (not measured) | 103 | 85.8 | 38 | 88.4 | 65 | 84.4 | 0.786 |
| | Cough | 56 | 46.7 | 26 | 60.5 | 30 | 39.0 | **0.035** |
| | Sore throat | 53 | 44.2 | 18 | 41.9 | 35 | 45.5 | 0.848 |
| | Malaise | 77 | 64.2 | 37 | 86.0 | 40 | 51.9 | **<0.001** |
| | Coryza | 30 | 25.0 | 6 | 14.0 | 24 | 31.2 | 0.048 |
| | Anosmia/Ageusia | 20 | 16.7 | 14 | 32.6 | 6 | 7.8 | **<0.001** |
| | Myalgia | 46 | 38.3 | 25 | 58.1 | 21 | 27.3 | **0.002** |
| | Arthralgia | 35 | 29.2 | 20 | 46.5 | 15 | 19.5 | **0.003** |
| | Diarrhoea | 24 | 20.0 | 9 | 20.9 | 15 | 19.5 | 1.000 |
| | Nausea | 23 | 19.2 | 8 | 18.6 | 15 | 19.5 | 1.000 |
| | Vomiting | 8 | 6.7 | 3 | 7.0 | 5 | 6.5 | 1.000 [b] |
| | Headache | 41 | 34.2 | 21 | 48.8 | 20 | 26.0 | **0.016** |
| | Anxiety | 16 | 13.3 | 5 | 11.6 | 11 | 14.3 | 0.785 [b] |

[a] Chi-square tests.

[b] Fisher's Exact test.

Statistically significant differences (p < 0.05) are marked in bold.

Comparing the two observed tests, confirmation of COVID-19 using the STANDARD Q COVID-19 Ag Test was significantly associated (0.028) with the shorter period (4.4±2.48) between the disease onset and the day of sampling compared to the RT-qPCR (8.09±61.57). On the other hand, differences between the two tests among laboratory-confirmed cases regarding age and gender were not statistically significant (p>0.05) (Table 3).

Finally, we have also compared the rapid Ag test results with the results of the quantitative RT-qPCR. Based on 43 RT-qPCR positive results, a median Cycle threshold (Ct) value was 26.5 (range: 12–41). The 25 concordant positive samples (positive results obtained both with RT-qPCR and STANDARD Q COVID-19 Ag Test) had a median Ct of 25 (range: 12–36), equivalent to a median of 676,851 copies/mL, whereas the median Ct of the 18 discordant samples (positive RT-qPCR with negative Ag rapid test) was 33 (range: 21–41), corresponding to a median of 557 copies/mL. During the first five days of disease, when the STANDARD Q COVID-19 Ag Test had the best performance, median number (676,851 copy number/mL) of copies was 177 time higher than the median number (3833 copy number/mL) after this period of disease (Table 4).

**Table 2. Validation of the STANDARD Q COVID-19 Ag Test by gender, age groups and days of signs/symptoms onset.**

| Characteristics | | Se [a] % (95% CI) | Sp [b] % (95% CI) | PPV % (95% CI) | NPV % (95% CI) | Accuracy (95% CI) | Kappa (95% CI) |
|---|---|---|---|---|---|---|---|
| **Gender** | Male | 57.1 (37.2–75.5) | 100 (-) | 100 (-) | 74.5 (65.5–81.7) | 0.81 (0.69–0.89) | 0.597 (0.409–0.785) |
| | Female | 60.0 (32.3–83.7) | 100 (-) | 100 (-) | 87.5 (79.0–92.9) | 0.89 (0.78–0.96) | 0.689 (0.465–0.913) |
| **Age group** | 14–30 | 66.7 (9.4–99.2) | 100 (-) | 100 (-) | 90.9 (66.9–98.0) | 0.92 (0.64–0.99) | 0.755 (0.307–1.000) |
| | 31–64 | 65.6 (46.8–81.4) | 100 (-) | 100 (-) | 78.0 (68.7–85.1) | 0.85 (0.74–0.92) | 0.677 (0.511–0.843) |
| | ≥65 | 25.0 (3.2–65.1) | 100 (-) | 100 (-) | 82.4 (75.8–87.4) | 0.83 (0.67–0.94) | 0.341 (-0.021–0.703) |
| **Day/s of signs/symptoms onset** | First | 100 (-) | 100 (-) | 100 (-) | 100 (-) | 1.00 (-) | 1.000 (-) |
| | Second | 83.3 (35.9–99.6) | 100 (-) | 100 (-) | 90.0 (60.1–98.2) | 0.93 (0.68–0.99) | 0.857 (0.589–1.000) |
| | Third | 66.7 (22.3–95.7) | 100 (-) | 100 (-) | 85.7 (22.3–95.7) | 0.89 (0.66–0.99) | 0.727 (0.384–1.000) |
| | Fourth | 71.4 (29.0–96.3) | 100 (-) | 100 (-) | 71.4 (43.7–89.0) | 0.83 (0.52–0.98) | 0.676 (0.288–1.000) |
| | Fifth | 100 (-) | 100 (-) | 100 (-) | 100 (-) | 1.00 (-) | 1.00 (-) |
| | **Subtotal (1–5)** | **79.2 (57.9–92.9)** | **100 (-)** | **100 (-)** | **88.6 (78.2–94.5)** | **0.92 (0.82–0.97)** | **0.825 (0.680–0.970)** |
| | 6–10 | 55.6 (21.2–86.3) | 100 (-) | 100 (-) | 73.3 (57.0–85.1) | 0.80 (0.56–0.94) | 0.579 (0.244–0.914) |
| | 11–15 | 50.0 (1.3–98.7) | 100 (-) | 100 (-) | 85.7 (60.0–96.0) | 0.88 (0.47–0.99) | 0.600 (-0.072–1.000) |
| | ≥16 | 0 (-) | 100 (-) | NA | 72.4 (72.4–72.4) | 0.72 (52.8–87.3) | NA |
| **Overall** | | **58.1 (42.1–73.0)** | **100 (-)** | **100 (-)** | **81.1 (75.1–85.9)** | **0.85 (0.78–0.91)** | **0.641 (0.498–0.783)** |

[a] Sensitivity

[b] Specificity; NA-not applicable (when the sensitivity is zero).

## Discussion

It seems that the actual number of COVID-19 cases in many countries is much higher than reported due to the limited testing since the beginning of the COVID-19 pandemic [2, 15]. Use of RT-qPCR test kits for laboratory confirmation of SARS-CoV-2 is the gold-standard for the diagnosis of COVID-19. However, this technique requires accredited medical laboratories, with advanced analytical instruments and trained personnel [2, 3, 6–8, 16]. Bearing this in mind, the low-cost rapid antigen COVID-19 test with a good diagnostic performance represents a global healthcare necessity. According to WHO, "The rapid diagnostic test refers to decentralized testing that is performed by a minimally trained healthcare professional near a patient and outside of central laboratory testing" [17].

To the best of our knowledge, this is the first study on the validation of the STANDARD Q COVID-19 Ag Test tests for SARS-CoV-2 infection for the clinical use in Serbia. Additionally,

**Table 3. Comparison between RT-qPCR and STANDARD Q COVID-19 Ag Test laboratory confirmed cases of SARS-CoV-2 infections.**

| Characteristics | | RT-qPCR positive | STANDARD Q COVID-19 Ag Test positive | P value |
|---|---|---|---|---|
| **Age (years)** | Mean ± SD | 47.11±14.61 | 44.29±11.57 | 0.453 [a] |
| | Median (IQR) | 48 (35–57) | 44 (34–52) | |
| **Gender** | Male | 28 | 16 | 0.865 [b] |
| | Female | 15 | 9 | |
| **Days of symptoms onset to testing (Mean ± SD)** | | 8.09±61.57 | 4.4±2.48 | **0.028 [a]** |

[a] ANOVA analysis of variance/Kruskal±Wallis H test.

[b] Chi-square test.

Statistically significant difference ($p < 0.05$) is marked in bold.

**Table 4. Median cycle threshold and quantification of PCR and STANDARD Q COVID-19 Ag Test results.**

| Characteristics | Median cycle threshold (Ct) value | | Quantity (copy number/mL) | |
|---|---|---|---|---|
| | Median | Range | Median | Range |
| Positive PCR (n = 43) | 26.5 | 12–41 | 110,298 | 42–163,790,516 |
| Positive PCR and STANDARD Q COVID-19 Ag Test (n = 25) | 25.0 | 12–36 | 676,851 | 196–163,790,516 |
| Positive PCR with negative STANDARD Q COVID-19 Ag Test (n = 18) | 33.0 | 21–41 | 557 | 42–766,740 |
| First five days post symptom onset | 25.0 | 12–38 | 676,851 | 42–163,790,516 |
| ≥ 6 days post symptom onset | 32.0 | 21–41 | 3833 | 170–766,740 |

unlike other studies [18–26], this is the first study that provides the evidence of STANDARD Q COVID-19 Ag Test accuracy considering the days from symptom onset among patients who did not require hospitalization.

We determined the diagnostic performance of the STANDARD Q COVID-19 Ag Test for detecting SARS-CoV-2 virus in the upper respiratory tract samples, and compared the results with RT-qPCR test. Only patients with mild or moderate clinical signs and symptoms of COVID-19 were included in the study.

Here we presented evidence that all patients with symptoms related to SARS-CoV-2 infection who had positive STANDARD Q COVID-19 Ag Test were adequately classified as true positive, i.e. there were no respondents with STANDARD Q COVID-19 Ag Test positive who had RT-qPCR negative test. Owning to its Sp and PPV value of 100% and accuracy of 0.85, subjects in our study with a positive STANDARD Q COVID-19 Ag Test could receive immediate care and be isolated while those with STANDARD Q COVID-19 Ag Test negative results needed further RT-qPCR examination. Bearing this in mind, it seems that this STANDARD Q COVID-19 Ag Test has the promising diagnostic capacity for replacement of the RT-qPCR, especially during the outbreak, when there is an increasing incidence trend of COVID-19 cases in the population. This could also accelerate clinical decision making in majority of suspected patients, which is in line with the strategy to stop the current spread of infection in the community. It is worth noticing that validation of this test demonstrated higher diagnostic performance if it was used during the early phase of the illness (preferably in the first five days following symptoms onset) when diagnostic accuracy ranged between 0.83 and 1 (average 0.92, 95% CI 0.82–0.97).

Our results also showed that laboratory confirmation of SARS-CoV-2 using this STANDARD Q COVID-19 Ag Test was slightly more precise in younger compared to the oldest subjects. Yet, this finding must be interpreted with caution, especially considering the time from the symptom onset. Namely, the sensitivity of the STANDARD Q COVID-19 Ag Test in patients aged ≥65 years was only 25%, contrary to 14–30 and 31–64 age groups where it was about 66%. These results indicate that only one out of four elderly patients with signs/symptoms related to COVID-19 had a laboratory confirmation of SARS-CoV-2 infection using the STANDARD Q COVID-19 Ag Test. Given that the highest diagnostic accuracy of the rapid test was observed in the first five days after symptom onset, the observed discrepancy between the age groups of participants could be ascribed to a different average time elapsed between disease onset and testing, as it was 12 days among elderly patients was, compared to only 8 and 6 days in those aged 31–64 and 14–30 years, respectively,.

On the other hand, our findings indicate that the STANDARD Q COVID-19 Ag Test was less sensitive than RT-qPCR. Thus, negative results from the STANDARD Q COVID-19 Ag Test cannot exclude SARS-CoV-2 virus infection confidently and such results should be verified by further RT-qPCR testing. For this reason, we believe that the STANDARD Q COVID-19 Ag Test have the potential to replace the RT-qPCR test, but only to some extent [27].

Nevertheless, it is worth noticing that the average time periods from symptom onset to the testing for RT-qPCR and STANDARD Q COVID-19 Ag Test were 8 and 4 days, respectively.

It has been clearly shown that the probability of laboratory confirmation of COVID-19 using rapid antigen tests depends on the viral load in specimens [7, 22, 27, 28], and that the high loads of SARS-CoV-2 virus correlates with the very early respiratory symptomatic stage of COVID-19 [29, 30]. Although the RT-qPCR for SARS-CoV-2 virus can yield positive results as late as 83 days after symptom onset, detection of viral RNA by RT-qPCR is not necessarily associated to infectiousness [31]. In fact, viral culture from PCR positive upper respiratory tract samples has been rarely positive after the ninth day of illness [31–33]. Consistent with the findings of other authors [34], we here presented strong evidence that the probability of positive STANDARD Q COVID-19 Ag Test matched with the period of the highest viral load obtained by RT-qPCR test. Among STANDARD Q COVID-19 Ag Test positive tests, 71.4% of them had a high viral load (Ct<25). Median Ct for concordant RT-qPCR and STANDARD Q COVID-19 Ag Test positive results was 25, i.e. median copy number/mL was 676,851, which coincided with the results in patients tested within the first five days of illness. In contrast, low viral load (557 copy number/mL) was associated with the negative STANDARD Q COVID-19 Ag Test despite a potentially positive RT-qPCR test. Taking into account all these findings, it seems reasonable to use the STANDARD Q COVID-19 Ag Test at the healthcare settings where patients are admitted in the first five days after the onset of symptoms, and probably also among close contacts of index cases in families and collectives. On the other hand, based on the results of this validation, we believe that this test is not to be recommended for clinical laboratory practice at inpatient facilities where patients are admitted in the late stage of the illness, which is consistent with the recommendation of WHO [10]. Moreover, due to low viral loads, results of nasopharyngeal specimens in those patients could be false negative not only after STANDARD Q COVID-19 Ag Test use, but also after the testing using RT-qPCR [10, 33, 34].

Our study had some limitations. First, this study was performed during the descendent phase of the epidemiological curve in Vojvodina when the prevalence of positivity in the population was ≤5% [35]. We also did not collect specimens from asymptomatic patients. However, considering the performance of the STANDARD Q COVID-19 Ag Test, we believe that this test should not be used in the asymptomatic population (low viral shedding) in a low prevalence setting. Additionally, we believe that further research conducted on a bigger scale, including larger population sample will provide more reliable values of the sensitivity of the STANDARD Q COVID-19 Ag Test. Second, as a result of this study being conducted during late summer in Serbia with a low circulation of other respiratory viruses, it is possible that the obtained performance of the STANDARD Q COVID-19 Ag Test could be different in other periods of the year. Third, although we had organized training among health care providers on adequate way of sampling and the samples handling procedures before the study, it is possible that some of the swabs were inadequately sampled, which could have potentially resulted in false negative results.

In conclusion, results of this study provide the evidence that the STANDARD Q COVID-19 Ag Test with specificity and PPV of 100% represents a promising tool for the rapid diagnostic ("near patient" use) of SARS-CoV-2 infection in the population with high prevalence of COVID-19. The STANDARD Q COVID-19 Ag Test is preferable to be used during the first five days of illness when the accuracy of this test reached at 0.92 and Kappa coefficient showed a strong level (0.825) of agreement [14] with RT-qPCR test. In addition to its high diagnostic accuracy for confirming COVID-19 infection, this easy-to-use antigen SARS-CoV-2 test can be used to test large number of samples in a short period of time. However, due to the possibility of false negative STANDARD Q COVID-19 Ag Test results, we also recommend that the

decision on further testing should be based on the clinical presentation of the patient [36] and, if possible, completed by RT-qPCR or IgM serology testing [37].

## Acknowledgments

We thank the all health care workers from Health Centre Novi Sad and Clinical Centre of Vojvodina, Department for Infectious Diseases, Novi Sad as well as the clinical laboratory technicians at the Centre for Virusology of Institute of Public Health of Vojvodina, Novi Sad who participated in this project.

A special thanks goes to Dr Jasmina Boban (Department for Radiology, Faculty of Medicine, University of Novi Sad, Novi Sad, Serbia), and prof. Miloš Marković (Department of Immunology, Institute of Microbiology and Immunology, Faculty of Medicine University of Belgrade, Belgrade, Serbia) for the revision of the manuscript. We gratefully acknowledge to prof. Edita Stokić for her support during the study.

## Author Contributions

**Conceptualization:** Mioljub Ristić, Velibor Čabarkapa, Vesna Turkulov, Vladimir Petrović.

**Data curation:** Nataša Nikolić, Velibor Čabarkapa, Vesna Turkulov.

**Formal analysis:** Nataša Nikolić, Velibor Čabarkapa.

**Funding acquisition:** Velibor Čabarkapa.

**Investigation:** Mioljub Ristić, Nataša Nikolić, Vesna Turkulov.

**Methodology:** Mioljub Ristić, Nataša Nikolić, Vesna Turkulov, Vladimir Petrović.

**Project administration:** Mioljub Ristić, Velibor Čabarkapa.

**Resources:** Mioljub Ristić, Velibor Čabarkapa, Vesna Turkulov.

**Supervision:** Vesna Turkulov.

**Validation:** Mioljub Ristić.

**Visualization:** Mioljub Ristić.

**Writing – original draft:** Mioljub Ristić.

**Writing – review & editing:** Mioljub Ristić, Vladimir Petrović.

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
