## [Decision Letter · Decision Letter 0]

16 Dec 2020

PONE-D-20-34883

Validation of the COVID-19 Rapid Antigen Test in Vojvodina, Serbia

PLOS ONE

Dear Dr. Ristić,

Thank you for submitting your manuscript to PLOS ONE. After careful consideration, we feel that it has merit but does not fully meet PLOS ONE’s publication criteria as it currently stands. Therefore, we invite you to submit a revised version of the manuscript that addresses the points raised during the review process.

We look forward to receiving your revised manuscript.

Kind regards,

John Schieffelin, MD

Academic Editor

PLOS ONE

Journal Requirements:

2. We note that patients provided oral consent to have a swab taken. Please state whether the IRB or ethics committee waived the requirement for informed consent to participate in your specific study. If patients provided informed written consent to have data/samples from their medical records used in research, please include this information.

3. In the ethics statement in the manuscript and in the online submission form, please provide additional information about the patient records/samples used in your retrospective study, including the date range (month and year) during which patients' medical records/samples were accessed.

"This work is a part of the research that was supported by Provincial Secretariat for Higher

Education and Scientific Research grant number 142-451-3072/2020-03."

Reviewers' comments:

Reviewer's Responses to Questions

**Comments to the Author**

1. Is the manuscript technically sound, and do the data support the conclusions?

Reviewer #1: Yes

Reviewer #2: No

2. Has the statistical analysis been performed appropriately and rigorously? 

Reviewer #1: Yes

Reviewer #2: No

3. Have the authors made all data underlying the findings in their manuscript fully available?

Reviewer #1: Yes

Reviewer #2: No

4. Is the manuscript presented in an intelligible fashion and written in standard English?

Reviewer #1: Yes

Reviewer #2: No

5. Review Comments to the Author

Reviewer #1: This is a well presented manuscript. The authors do a good job of addressing study limitations and caveats. One point is that the viral transport medium (VTM) used for qPCR is likely not optimal for performing an antigen test. The authors should specify whether or not the any of the swabs were collected in VTM with guanidine or other disruptive agent as this could decrease antigen detection. The authors should also mention the manufacturer of the test in the abstract as there are several and performance will vary. The sample size is limited, which should also be mentioned in the abstract.

Reviewer #2: General Comments:

The authors evaluated the performance of the STANDARD Q COVID-19 Ag Test among symptomatic patients who presented to health care facilities in Novi Sad, Serbia. Performance of the rapid antigen test was compared with qRT-PCR (the gold-standard diagnostic test recommended by WHO). The authors observed a strong agreement between performance of the rapid antigen test and qRT-PCR.

1. The author acknowledged the importance and novelty of such a validation in their setting and globally; however, it would have been helpful for the authors to set out a well- structured study design, with a well-defined and described study population, and selection criteria. There were major methodological weaknesses making the study design prone to a lot of biases, and hence, interpretation of the results.

2. Overall, there were many grammatical and scientific-writing errors.

3. It would have been helpful for the authors to use the standard manufacturer’s name (STANDARD Q COVID-19 Ag Test) of the rapid antigen test used for this validation.

Specific Comments:

1. Abstract:

a. The result and conclusion could benefit from review of grammatical and scientific-writing errors.

2. Introduction:

a. It would be helpful for the authors to review grammatical and scientific-writing errors.

b. Line 88: It would be helpful for the authors to clearly explain or edit sentence stating “no strong evidence to determine the usefulness of these tests in clinical practice”. What is being referred to as ‘strong evidence’?

c. Lines 93-94: The use of the words “potential partial replacement of qRT-PCR” appears misleading. The relevance of such a statement could be spelt out, otherwise it is debatable. Most studies have shown that the use of COVID-19 rapid antigen tests, as with many other rapid antigen tests is complementary to qRT-PCR.

3. Study design:

a. Lines 97- 99: It would be helpful for the authors to clearly state a specific study design, and to explain what is meant by “retrospective analysis of prospective collected data”. Was it a prospective or retrospective study?

b. It would be helpful for the authors to justify reasons for the short duration of the study which is a possible limitation for the generalization of the study; an inclusion of the COVID-19 prevalence would have been an added advantage in interpreting results from PPV and NPV.

c. Lines 102-104: The relevance of these sentence should be spelt out or reviewed.

d. Lines 106 -107: Inclusion and exclusion criteria should be rigid, specific and clearly stated. Information on these lines appear contrary to Line 100; Including all symptomatic patients could mean both in-patients and outpatients; COVID-19 symptoms are non-specific; hence it would have been helpful for the authors to state the criteria used by these facilities to define who a case suspected of COVID-19 is; and also, using a non-specific age group makes such a study difficult to reproduce.

e. Lines 107-109: The authors alluded that “physicians through face-face structured interviews on the day of admission interviewed all included participants”. It would be helpful for the authors to clarify what they meant by ‘day of admission’, as this sentence appears contradictory to Line 100.

f. Lines 118-119: The authors made mention of “previously trained medical staff”; the text could benefit from an explanation.

g. Lines 120-122: The authors alluded to “available tubes containing a specific viral transport medium with antifungal and antibiotic supplements”; what was the role of the antifungal and antibiotic supplements being referred to here?

h. Lines 122-126: It would be helpful for the authors to cite a reference for the laboratory protocol used.

i. Lines 128-152: Cite reference(s); this text could be shortened.

j. Lines 156-157: Cite reference(s).

4. Data Analysis:

a. Lines 174-176: The relevance of the broad classification of statistical analyses used should be spelt out, otherwise it would be helpful for the authors to indicate specific statistical analyses used.

b. Lines 177-178: The authors should review the scientific-writing of statistical tests. For example, it is commonly written Kruskal Wallis test or Kruskal-Wallis ANOVA rather than ANOVA-Kruskal±Wallis H-test; and Fisher’s Exact test rather than Fisher exact test. It would be helpful for the authors to also indicate that Kruskal-Wallis ANOVA was used for ordinal scale variables.

c. Lines 184-186: It would be helpful for the authors to clarify whether the accuracy and Kappa values used were derived from validation done by the manufacturer.

d. How was sample size determined?

e. Any consideration for missing data?

5. Ethical Considerations:

a. Lines 190-191: Why was verbal informed consent obtained only at the moment of sample collection? It would be helpful for the authors to explain why informed consent was not sought for the overall conduct of the research which also include, but not limited to data and sample collection.

6. Results:

a. It would be helpful for the authors to review the scientific-writing in this section.

b. Line 196: The relevance of the information should be spelt out as this sentence appears contrary to Line 100 stating only outpatients were enrolled in the study.

c. Lines 196-198: The relevance of the information should be spelt out, otherwise its inclusion is questionable.

d. It would be helpful for the authors to explain why fever was not measured.

e. Lines 201: The relevance of the information in line should be spelt out.

f. Lines 203-204: It would be helpful for the authors to explain the relevance of both a mean period of 9.4 days and a median time of 5.

g. Lines 206-207: The relevance of the information in these lines should be spelt out.

h. Lines 225-227: It would be helpful for the authors to explain the relevance of the values in the brackets.

i. What is the overall observed period in Line 229 referring to?

7. Discussion:

a. It would be helpful for the authors to review the grammar and scientific-writing in the section, and by extension interpretation of the results.

b. Line 257: The relevance of the information could be spelt out, otherwise its inclusion in this section is debatable

c. Lines 266-268: These lines appear contradictory to Lines 107-109 and Line 196. Were patients who presented with symptoms requiring admission excluded because they it was an exclusion criterion, or no patient with symptoms needed admission?

d. Lines 271-272: These lines appear contradictory to Line 106 stating all symptomatic patients were included. It would be helpful for the authors to clarify.

e. Cite reference(s) for Lines 304-306.

f. Lines 324: Cite reference stating the prevalence of positivity

g. Was there an explanation for participants presenting to facilities and being tested after ≥ 16 days from onset of signs/symptoms?

8. Tables:

a. Table 1: It would be helpful for the authors to consider using figures for “days from symptoms onset”; for example, 0-5 days for the “first five days”.

b. Line 215: It would be helpful for the authors to differentiate P values derived from Fisher’s Exact test or chi-square test

c. Table 2: It would be helpful for the authors to explain the meaning of NA underneath the table;

d. Table 2: Lines 239-240: The title of the table does not reflect the findings; the authors should consider replacing “Similarities and differences” with “Comparison”.

9. Figures:

a. Figure 2: The authors should consider indicating what unit is the period between symptom onset and laboratory testing on the Y-axis measured in?

6. PLOS authors have the option to publish the peer review history of their article (what does this mean?). If published, this will include your full peer review and any attached files.

Reviewer #1: No

Reviewer #2: **Yes: **Robert J. Samuels

---

## [Author Response · Author response to Decision Letter 0]

19 Jan 2021

Response to Reviewers

IMPORTANT!

Comments to the Author are shown in regular fonts, while the comments to the academic Editor are in the italic fonts.

In accordance with your suggestion and the suggestions of the Reviewers, we added:

Fig 2 (We added (days) on the Y-axis as Reviewer #2 suggested).

Reviewer #1: This is a well presented manuscript. The authors do a good job of addressing study limitations and caveats. One point is that the viral transport medium (VTM) used for qPCR is likely not optimal for performing an antigen test. The authors should specify whether or not the any of the swabs were collected in VTM with guanidine or other disruptive agent as this could decrease antigen detection. The authors should also mention the manufacturer of the test in the abstract as there are several and performance will vary. The sample size is limited, which should also be mentioned in the abstract.

Thank you very much for those suggestions. As we explained in the Materials and methods section of the paper, our analysis of STANDARD Q COVID-19 Ag Test was performed in the “while you wait” manner - without the need for transportation of the sample to the laboratory. Thus, VTM was not used for testing with STANDARD Q COVID-19 Ag Test. However, according to the suggestion of the Reviewer #1, we added a more precise description of the procedure in the appropriate section of the paper.

In the sample collection and laboratory testing section (STANDARD Q COVID-19 Ag Test subsection) we mentioned that the manufacturer of STANDARD Q COVID-19 Ag Test was SD Biosensor, Gyeonggi-do, South Korea. This information is now also added to the Abstract section as well as the remark on the study sample size.

Reviewer #2:

General Comments:

The authors evaluated the performance of the STANDARD Q COVID-19 Ag Test among symptomatic patients who presented to health care facilities in Novi Sad, Serbia. Performance of the rapid antigen test was compared with qRT-PCR (the gold-standard diagnostic test recommended by WHO). The authors observed a strong agreement between performance of the rapid antigen test and qRT-PCR.

1. The author acknowledged the importance and novelty of such a validation in their setting and globally; however, it would have been helpful for the authors to set out a well- structured study design, with a well-defined and described study population, and selection criteria. There were major methodological weaknesses making the study design prone to a lot of biases, and hence, interpretation of the results.

2. Overall, there were many grammatical and scientific-writing errors.

3. It would have been helpful for the authors to use the standard manufacturer’s name (STANDARD Q COVID-19 Ag Test) of the rapid antigen test used for this validation.

Thank you for these suggestions. We tried to properly correct all the errors you mentioned. Our methodology was similar to other research studies that we listed in the Reference section. In addition, we replaced “AG-RDT”, “COVID-19 Ag rapid diagnostic test” and “Rapid Ag-RDT for COVID-19” with “STANDARD Q COVID-19 Ag Test” throughout the manuscript body. Finally, in order to better clarfy our study design, the study population and selection criteria have been explained in more details in the revised version of the manuscript.

Finally, the copy editing and English language, spelling and grammar check were performed throughout the paper.

We want to highlight that copy editing and English language, spelling and grammar check were performed by Dr Jasmina Boban (Department for Radiology, Faculty of Medicine, University of Novi Sad, Novi Sad, Serbia). Dr Jasmina Boban previously wrote the following papers in correct English language (four of all listed articles previously published in the PLoS ONE):

- Performance of the new clinical case definitions of pertussis in pertussis suspected infection and other diagnoses similar to pertussis. PLoS ONE. 2018; 13(9): e0204103. doi: 10.1371/journal.pone.0204103.

- Sero-epidemiological study in prediction of the risk groups for measles outbreaks in Vojvodina, Serbia. PLoS ONE. 2019;14(5): e0216219.

- Declining seroprevalence of hepatitis A in Vojvodina, Serbia. PLoS ONE. 2019;14(6): e0217176.

- Seroepidemiological study of rubella in Vojvodina, Serbia: 24 years after the introduction of the MMR vaccine in the national immunization programme. PLoS ONE. 2020;15(1): e0227413.

- Differentiation of Breast Lesions and Distinguishing Their Histological Subtypes Using Diffusion-Weighted Imaging and ADC Values. FRONTIERS IN ONCOLOGY. 2020.

- Neurometabolic Remodeling in Chronic Hiv Infection: a Five-Year Follow-up Multi-Voxel Mrs Study. SCIENTIFIC REPORTS. 2019.

- Thalamic volume loss as an early sign of amnestic mild cognitive impairment. JOURNAL OF CLINICAL NEUROSCIENCE. 2019; 68:168-173.

- Susceptibility-Weighted MR Imaging Hypointense Rim in Progressive Multifocal Leukoencephalopathy: The End Point of Neuroinflammation and a Potential Outcome Predictor. AMERICAN JOURNAL OF NEURORADIOLOGY. 2019.

- The role of TNF-alpha superfamily members in immunopathogenesis of sepsis. CYTOKINE. 2018; 111:125-130.

- Apparent diffusion coefficient reproducibility in brain tumors measured on 1.5 and 3 T clinical scanners: A pilot study. EUROPEAN JOURNAL OF RADIOLOGY. 2018; 108:249-253.

- Early Introduction of cART Reverses Brain Aging Pattern in Well-Controlled HIV Infection: A Comparative MR Spectroscopy Study. FRONTIERS IN AGING NEUROSCIENCE. 2018.

- Executive Functions Rating Scale and Neurobiochemical Profile in HIV-Positive Individuals. FRONTIERS IN PSYCHOLOGY, (2018).

- Complement component consumption in sepsis correlates better with hemostatic system parameters than with inflammatory biomarkers. THROMBOSIS RESEARCH. 2018; 170:126-132.

-APRIL and sTACI could be predictors of multiorgan dysfunction syndrome in sepsis. Virulence. 2018; 9(1):946-953. doi: 10.1080/21505594.2018.1462636.

-Leptomeningeal form of Immunoglobulin G4-related hypertrophic meningitis with perivascular spread: a case report and review of the literature. Neuroradiology. 2018. doi: 10.1007/s00234-018-2028-y.

-Basal ganglia shrinkage without remarkable hippocampal atrophy in chronic aviremic HIV-positive patients. J Neurovirol. 2018. doi: 10.1007/s13365-018-0635-3.

-A prominent lactate peak as a potential key magnetic resonance spectroscopy (MRS) feature of progressive multifocal leukoencephalopathy (PML): Spectrum pattern observed in three patients. Bosn J Basic Med Sci. 2017; 17(4):349-354. doi: 10.17305/bjbms.2017.2092.

-HIV-associated neurodegeneration and neuroimmunity: multivoxel MR spectroscopy study in drug-naive and treated patients. Eur Radiol. 2017; 27(10):4218-4236. doi: 10.1007/s00330-017-4772-5.

-Proton Chemical Shift Imaging Study of the Combined Antiretroviral Therapy Impact on Neurometabolic Parameters in Chronic HIV Infection. AJNR Am J Neuroradiol. 2017; 38(6):1122-1129. doi: 10.3174/ajnr.A5160.

-Bilateral bloody nipple discharge in a male infant: sonographic findings and proposed diagnostic approach. J Pediatr Endocrinol Metab. 2012; 25(1-2):163-4. doi: 10.1515/jpem-2011-0460.

Specific Comments:

1. Abstract:

a. The result and conclusion could benefit from review of grammatical and scientific-writing errors.

The abstract has been substantially revised, and copy editing and English language, spelling and grammar check were performed

2. Introduction:

a. It would be helpful for the authors to review grammatical and scientific-writing errors.

The introduction section has also been revised and copy editing and English language, spelling and grammar check were performed.

b. Line 88: It would be helpful for the authors to clearly explain or edit sentence stating “no strong evidence to determine the usefulness of these tests in clinical practice”. What is being referred to as ‘strong evidence’?

Thank you for this suggestion. We stated that “So far, there is no strong evidence available to determine the usefulness of these tests in clinical practice” reflecting that at the time when the paper was drafted, there was a lack of evidence for the level of clinical performance of this rapid antigen test. Supporting this fact, , Cochrane Database of Systematic Reviews (Dinnes J, Deeks JJ, Adriano A, Berhane S, Davenport C, Dittrich S, et al. (Cochrane COVID-19 Diagnostic Test Accuracy Group). Rapid, point-of-care antigen and molecular-based tests for diagnosis of SARS-CoV-2 infection. Cochrane Database Syst Rev. 2020;8:CD013705. doi: 10.1002/14651858) highlighted that “Antigen tests Sensitivity varied considerably across studies (from 0% to 94%): the average sensitivity was 56.2% (95% CI 29.5 to 79.8%) and average specificity was 99.5% (95% CI 98.1% to 99.9%; based on 8 evaluations in 5 studies on 943 samples). Data for individual antigen tests were limited with no more than two studies for any test.” Additionally, the article: Mak GC, Cheng PK, Lau SS, Wong KK, Lau CS, Lam ET, et al. Evaluation of rapid antigen test for detection of SARS-CoV-2 virus. J Clin Virol. 2020;129:104500. doi: 10.1016/j.jcv.2020.104500. stated: “However, according to WHO, the role of RAD tests for antigen detection for SARS−COV-2 needs to be evaluated and is not recommended for clinical diagnosis”, and “Although our data indicated that RAD test was capable of detecting SARS-CoV-2 virus in NPA & TS, NPS & TS, sputum and throat saliva with different sensitivities, this method was less sensitive than RT-PCR. Consequently, the negative results from this RAD method cannot exclude SARS-CoV-2 virus infection confidently and thus results should be verified by further RT-PCR testing.” The sentence in our paper was based on abovementioned facts.

c. Lines 93-94: The use of the words “potential partial replacement of qRT-PCR” appears misleading. The relevance of such a statement could be spelt out, otherwise it is debatable. Most studies have shown that the use of COVID-19 rapid antigen tests, as with many other rapid antigen tests is complementary to qRT-PCR.

Although we value this remark, we can not absolutely agree with it. Namely, rapid antigen tests are not absolutely complementary with results of RT-qPCR, as mentioned above in the Cochrane Database of Systematic Reviews. Consistently, our results showed that patients who had positive results obtained on rapid antigen test afterwards presented with positive RT-qPCR test. However, some of the patients with negative rapid antigen test also afterwards presented with positive RT-qPCR. More precisely, during the first five days of illness, around 20% of patients who had a negative rapid antigen test presented with positive RT-qPCR test. Based on this observation, we stated the “potential partial replacement of qRT-PCR”. Again, we lean on the Cochrane Database of Systematic Reviews that highlighted: “Where people are asymptomatic but are being tested on the basis of epidemiological risk factors, such as exposure to someone with confirmed SARS‐CoV‐2, no prior tests will have been conducted.”; “Point‐of‐care tests potentially have a role either as a replacement for RT‐PCR (if sufficiently accurate), or as a means of triaging and rapid management (quarantine or treatment, or both), with confirmatory RT‐PCR testing for negative results”.

3. Study design:

a. Lines 97- 99: It would be helpful for the authors to clearly state a specific study design, and to explain what is meant by “retrospective analysis of prospective collected data”. Was it a prospective or retrospective study?

Thank you for this interesting suggestion. Similar to methodology used in previously published paper in the PLoS ONE (Performance of the new clinical case definitions of pertussis in pertussis suspected infection and other diagnoses similar to pertussis. PLoS ONE. 2018; 13(9): e0204103. doi: 10.1371/journal.pone.0204103), the data was collected in the prospective manner, while the analysis was retrospective. However, in order to avoid any possible confusion, the word “retrospective” has been removed in the revised version of the manuscript.

b. It would be helpful for the authors to justify reasons for the short duration of the study which is a possible limitation for the generalization of the study; an inclusion of the COVID-19 prevalence would have been an added advantage in interpreting results from PPV and NPV.

Thank you for this helpful suggestion. However, we explained that the main reason for the short duration of the study was the previous deadline given by the project which we conducted. In addition, in the Limitations, we highlighted that this study was performed when the prevalence of the COVID-19 positive cases was ≥5% and that we are aware that a study with a bigger sample would probably improve the sensitivity of this rapid antigen test.

c. Lines 102-104: The relevance of these sentence should be spelt out or reviewed.

After consultations and literature review, we reviewed the sentence.

d. Lines 106 -107: Inclusion and exclusion criteria should be rigid, specific and clearly stated. Information on these lines appear contrary to Line 100; Including all symptomatic patients could mean both in-patients and outpatients; COVID-19 symptoms are non-specific; hence it would have been helpful for the authors to state the criteria used by these facilities to define who a case suspected of COVID-19 is; and also, using a non-specific age group makes such a study difficult to reproduce.

Thank you very much for this suggestion. Generally, due to the test characteristics which was not recommended for use in asymptomatic patients, we included only symptomatic patients. Also, we did not include already hospitalized patients, however, there were patients who were tested before admitting to the tertiary health care setting (the ambulance (“red zone”) of Clinical Centre of Vojvodina, Department for Infectious Diseases). In addition, in present version of the paper, we added the signs/symptoms that were inclusion criteria for the testing. At enrolment, we imposed no limits regrading the age of participants. But, after finishing of the study, we stratified participants into three age groups: 14-30, 31-64, and ≥65 years of age. That was the reason for the statement that it was “A retrospective analysis of prospectively collected data”. Finally, after analysis of the data, we realized that most of the participants were aged 31-64 years, while there were no participants younger than 14 years of age. 

e. Lines 107-109: The authors alluded that “physicians through face-face structured interviews on the day of admission interviewed all included participants”. It would be helpful for the authors to clarify what they meant by ‘day of admission’, as this sentence appears contradictory to Line 100.

We added more precise information about the inclusion criteria in the paper and changed “day of admission” into “day of sampling”.

f. Lines 118-119: The authors made mention of “previously trained medical staff”; the text could benefit from an explanation.

Thank you for this suggestion. At the end of section “Study design” we stated “The education about adequate sampling and the samples handling procedures of all included physicians and nurses was conducted before the start of research and lasted for 10 days”. We believe that this explanation is clear and detailed. In addition, we have refered to “study design” subsection in the revised sentence in the present version of the text.

g. Lines 120-122: The authors alluded to “available tubes containing a specific viral transport medium with antifungal and antibiotic supplements”; what was the role of the antifungal and antibiotic supplements being referred to here?

Thank you very much for this. According to the WHO guidance, for transport of samples for viral detection, the use of viral transport medium (VTM) containing antifungal and antibiotic supplements is recommended (ref. 9: World Health Organization. Laboratory testing strategy recommendations for COVID-19: interim guidance. Interim guidance, 21 March (2020)). Antibiotics and antifungals in the VTM reduce the risk of bacterial and fungal contamination during the swab collections as well as to maintain the viability and virulence of collected samples. It is known that bacteria and fungi from the respiratory tract and other sites can disrupt viral particles’ viability and/or degrade DNA and RNA if allowed to proliferate (ref. Bradley Ford, Felix Lam, Johnathan Wilson, Majd Moubarak. Role of Viral Transport Media in Sustaining COVID-19 Testing. Medical Lab Management. October 2020.). If the reviewer feels it is important for the paper, we are ready to include this explanation in the appropriate section.

h. Lines 122-126: It would be helpful for the authors to cite a reference for the laboratory protocol used.

We strongly agree with your suggestion and we added reference. Also, all laboratory procedures were performed according to the manufacturer's instructions.

i. Lines 128-152: Cite reference(s); this text could be shortened.

Thank you for this very useful suggestion. However, the authors feel that it is very important to present these references at one place for the readers. . In this part of the paper, we explained in detail the laboratory procedures in case that other readers/researches compare their results with ours. Nevertheless, if the reviewer insists on shortentning this section, we are ready to move this part to the supplementary material. Other comparable studies also explained laboratory procedures without citation of special reference, and many of them highlighted that procedures were performed according to manufacturer's instructions. Regarding this, we suggest to look for the references 1,7,8,17-19,22,24,26,27,36 (section at the end of the paper before corrections in the paper).

j. Lines 156-157: Cite reference(s).

Please refer to the previous explanation. .

4. Data Analysis:

a. Lines 174-176: The relevance of the broad classification of statistical analyses used should be spelt out, otherwise it would be helpful for the authors to indicate specific statistical analyses used.

Thank you for noticing that. In additional text, we clarified which specific statistical analyses were used.

b. Lines 177-178: The authors should review the scientific-writing of statistical tests. For example, it is commonly written Kruskal Wallis test or Kruskal-Wallis ANOVA rather than ANOVA-Kruskal±Wallis H-test; and Fisher’s Exact test rather than Fisher exact test. It would be helpful for the authors to also indicate that Kruskal-Wallis ANOVA was used for ordinal scale variables.

We completely agree with your suggestions and, in accordance with this, we replaced the words. 

c. Lines 184-186: It would be helpful for the authors to clarify whether the accuracy and Kappa values used were derived from validation done by the manufacturer.

Thresholds of Accuracy and Kappa values were determined according to the reference standards (ref. 12-14 before corection in the paper). Since the manufacturer did not derive these values, we conducted thevalidation of this test.

d. How was sample size determined?

As we explained in Materials and methods section of the paper, we included the first 120 participants. Consistent with the methodology in many other studies (see reference section of the paper), in this phase of COVID-19 pandemic, it was necessary to both start and finish the research promptly. Taking into account above-mentioned facts, we are aware of these limitations we suggested conduction of a further research with a bigger sample (see Limitation of the paper).

e. Any consideration for missing data?

Taking into account the relatively small set of information (questions about sociodemographic features, as well as of questions related to the date of symptoms onset and questions about all mild or moderate clinical signs and symptoms related to SARS-CoV-2 infection), we are convinced that data obtained from our participants was collected without missing data.

5. Ethical Considerations:

a. Lines 190-191: Why was verbal informed consent obtained only at the moment of sample collection? It would be helpful for the authors to explain why informed consent was not sought for the overall conduct of the research which also include, but not limited to data and sample collection.

It is a very helpful suggestion. We agree that this section of our paper can potentially confusethe readers. So, we added the explanation according to the Reviewer #2 suggestion.

6. Results:

a. It would be helpful for the authors to review the scientific-writing in this section. 

Review of the scientific-writing check was performed and the whole section was revised

b. Line 196: The relevance of the information should be spelt out as this sentence appears contrary to Line 100 stating only outpatients were enrolled in the study.

We agree with this, and we removed confusing part of mentioned sentence.

c. Lines 196-198: The relevance of the information should be spelt out, otherwise its inclusion is questionable.

This is a very interesting suggestion, but we believe that these data provided information about relevant characteristics of included participants regarding their clinical characteristics as well as the timing of swabs taking; therefore, we suggested keeping this in the paper.

d. It would be helpful for the authors to explain why fever was not measured.

Primarily because of the intention for the rapid action (as fast as possible) at the health care settings, we decided to collect only oral information about fever. So, we honestly stated that the measurement of the fever was not performed at the spot. As you can see, the goal of our research was not the validation of sensitivity, specificity or accuracy for the signs/symptoms of the participants but the determination of rapid antigen test performance regarding the age, gender of participants as well as the days from signs/symptoms onset . After all, we believe that the absence of measured fever did not discredit the main results obtained by our research.

e. Lines 201: The relevance of the information in line should be spelt out.

In line 201 we stated “Fig 1. Prevalence of signs and symptoms among participants tested for COVID-19.” according to the proposition of the PLOS One.

f. Lines 203-204: It would be helpful for the authors to explain the relevance of both a mean period of 9.4 days and a median time of 5.

According to the previous good publishing practice in the PLOS One (see: Ristić M, Radosavljević B, Stojanović VD, Đilas M, Petrović V. Performance of the new clinical case definitions of pertussis in pertussis suspected infection and other diagnoses similar to pertussis. PLoS ONE. 2018; 13(9): e0204103; Ristić M, Milošević V, Medić S, Djekić Malbaša J, Rajčević S, Boban J, Petrović V. Sero-epidemiological study in prediction of the risk groups for measles outbreaks in Vojvodina, Serbia. PLoS ONE. 2019;14(5): e0216219; Medić S, Anastassopoulou C, Milošević V, Nataša D, Rajčević S, Ristić M, Petrović V. Declining seroprevalence of hepatitis A in Vojvodina, Serbia. PLoS ONE 2019;14(6): e0217176.), we stated mean and median periods in the Results section. However, in order to avoid potential misunderstanding, we specified mean (average) and median period (in days) between signs/symptom onset and day of swabs taking more precisely.

g. Lines 206-207: The relevance of the information in these lines should be spelt out.

In lines 206-207 we stated “Fig 2. Scatter plot of period between signs and symptoms onset and swab collection for laboratory testing for COVID-19 by age.” according to the proposition of the PLOS One and similar to the Fig 1 in the our previous published research in the PLOS One: Ristić M, Milošević V, Medić S, Djekić Malbaša J, Rajčević S, Boban J, Petrović V. Sero-epidemiological study in prediction of the risk groups for measles outbreaks in Vojvodina, Serbia. PLoS ONE. 2019;14(5): e0216219.

h. Lines 225-227: It would be helpful for the authors to explain the relevance of the values in the brackets.

Thank you for this very interesting suggestion. We corrected this sentence according to the suggestion of the Reviewer #2.

i. What is the overall observed period in Line 229 referring to?

Thank you for this suggestion. We corrected this sentence according to the suggestion of the Reviewer #2.

7. Discussion:

a. It would be helpful for the authors to review the grammar and scientific-writing in the section, and by extension interpretation of the results.

Thank you for this suggestion. The corrections were performed.

b. Line 257: The relevance of the information could be spelt out, otherwise its inclusion in this section is debatable.

This sentence was stated in the relevant literature which we listed.This information was published in PLoS ONE (Grant MC, Geoghegan L, Arbyn M, Mohammed Z, McGuinness L, Clarke EL, et al. The prevalence of symptoms in 24,410 adults infected by the novel coronavirus (SARS-CoV-2; COVID-19): A systematic review and meta-analysis of 148 studies from 9 countries. PLoS One. 2020;15(6):e0234765. doi: 10.1371/journal.pone.0234765. PMID: 32574165; PMCID: PMC7310678.) as a systematic review and meta-analysis of 148 studies from 9 countries (“Since the patients in the included studies are likely to have moderate-severe disease warranting hospitalisation and thus testing, it is likely that we over-estimate the true prevalence of symptoms in the population. Consequently, the use of symptoms alone to screening adults for SARS-CoV-2 infection is likely to miss a substantial number of infected individuals.”). Similar findings were reported by other authors:” Because of a sudden increased demand for confirmatory diagnostic testing, mildly affected and asymptomatic individuals have limited access to laboratory testing. As a result of such circumstances, the number of confirmed SARS-CoV-2 infections can significantly underestimate the actual number of cases” or “Generally, mildly affected or asymptomatic individuals are not screened. As a result, the number of confirmed SARS-CoV-2 infections is largely underestimated.” in: Verity R, Okell LC, Dorigatti I, Winskill P, Whittaker C, Imai N, et al. Estimates of the severity of coronavirus disease 2019: a model-based analysis. Lancet Infect Dis. 2020;20(6):669-677. doi: 10.1016/S1473-3099(20)30243-7. Erratum in: Lancet Infect Dis. 2020: Erratum in: Lancet Infect Dis. 2020: PMID: 32240634; PMCID: PMC7158570. So, we added mentioned references (ref. No 2, and 15) in the reference section.

c. Lines 266-268: These lines appear contradictory to Lines 107-109 and Line 196. Were patients who presented with symptoms requiring admission excluded because they it was an exclusion criterion, or no patient with symptoms needed admission?

We agree that these facts may be potentially confusing. Due to the mild or moderate clinical signs and symptoms of COVID-19 disease, none of our patients required hospitalization (as we stated in the Discussion section). Regarding additional text which we put in the Materials and methods section (Inclusion/Exclusion criteria subsection), the first sentence in the Results section was corrected.

d. Lines 271-272: These lines appear contradictory to Line 106 stating all symptomatic patients were included. It would be helpful for the authors to clarify.

Similar to the above explanation, we added data to the Materials and methods section (Inclusion/Exclusion criteria subsection). In our research all symptomatic patients who had one or more of the following signs/symptoms: fever, malaise, cough, sore throat, myalgia, arthralgia, headache, coryza, diarrhoea, nausea, anosmia or ageusia, vomiting were included.

e. Cite reference(s) for Lines 304-306.

We added.

f. Lines 324: Cite reference stating the prevalence of positivity.

We added.

g. Was there an explanation for participants presenting to facilities and being tested after ≥ 16 days from onset of signs/symptoms? 

Thank you for this question. No specific explanation exists for this fact, except that we wanted to perform validation of this rapid antigen test among patients in the early (first five days) as well as in the late phase of disease. In this way, we were able to compare performance of this test regarding the different phases of disease. We can confirm that in a total of 29 patients with onset of the symptoms ≥16 prior to swab collection a delayed medical examination was performed. 

8. Tables: 

a. Table 1: It would be helpful for the authors to consider using figures for “days from symptoms onset”; for example, 0-5 days for the “first five days”.

Thank you for this suggestion. We accepted it and corrected in the Table 1.

b. Line 215: It would be helpful for the authors to differentiate P values derived from Fisher’s Exact test or chi-square test.

The values were derived in accordance with biostatistics proposition (similar statistical consideration was presented in the Table 2 - article previously published in the PLOS ONE: Arau´jo LO, Nunes AMPB, Ferreira VM, Cardoso CW, Feitosa CA, Reis MG, et al. Clinical and epidemiological features of pertussis in Salvador, Brazil, 2011–2016. PLoS ONE. 2020; 15(9): e0238932. https://doi.org/10.1371/journal. pone.0238932 

c. Table 2: It would be helpful for the authors to explain the meaning of NA underneath the table.

Thank you for this. We added the explanation what NA means. In brief, when sensitivity is zero, then it is not possible to calculate PPV and Kappa (please look at below listed references: 

Altman DG, Machin D, Bryant TN, Gardner MJ (Eds) (2000) Statistics with confidence, 2nd ed. BMJ Books. Gardner IA, Greiner M (2006) Receiver-operating characteristic curves and likelihood ratios: improvements over traditional methods for the evaluation and application of veterinary clinical pathology tests. Veterinary Clinical Pathology 35:8-17. 

Griner PF, Mayewski RJ, Mushlin AI, Greenland P (1981) Selection and interpretation of diagnostic tests and procedures. Annals of Internal Medicine 94:555-600. 

Hanley JA, McNeil BJ (1982) The meaning and use of the area under a receiver operating characteristic (ROC) curve. Radiology 143:29-36. 

Mercaldo ND, Lau KF, Zhou XH (2007) Confidence intervals for predictive values with an emphasis to case-control studies. Statistics in Medicine 26:2170-2183. 

Metz CE (1978) Basic principles of ROC analysis. Seminars in Nuclear Medicine 8:283-298. 

Zhou XH, NA Obuchowski, DK McClish (2002) Statistical methods in diagnostic medicine. New York: Wiley. 

Zweig MH, Campbell G (1993) Receiver-operating characteristic (ROC) plots: a fundamental evaluation tool in clinical medicine. Clinical Chemistry 39:561-577.).

d. Table 2: Lines 239-240: The title of the table does not reflect the findings; the authors should consider replacing “Similarities and differences” with “Comparison”.

Since we feel the suggestion is actually associated with Table 3 (not Table 2), we agreed with this and corrected the name of Table 3. 

9. Figures:

a. Figure 2: The authors should consider indicating what unit is the period between symptom onset and laboratory testing on the Y-axis measured in?

Thank you for this suggestion. We added “days” on the Y-axis.

The authors feel that the paper is overall of a better quality after including the well-intended and helpful remarks obtained from the reviewers. 

Thank you for the consideration of our manuscript, we look forward to the next step of the paper revisions and hope for the positive outcome. 

Sincerely, 

Mioljub Ristić, MD, PhD

Centre for Disease Control and Prevention, Institute of Public Health of Vojvodina, Novi Sad, Serbia

Futoška 121, Novi Sad 21 000, Serbia

E-mail: mioljub.ristic@mf.uns.ac.rs

---

## [Decision Letter · Decision Letter 1]

10 Feb 2021

Validation of the COVID-19 Rapid Antigen Test in Vojvodina, Serbia

PONE-D-20-34883R1

Dear Dr. Ristić,

We’re pleased to inform you that your manuscript has been judged scientifically suitable for publication and will be formally accepted for publication once it meets all outstanding technical requirements.

Kind regards,

John Schieffelin, MD

Academic Editor

PLOS ONE

Additional Editor Comments (optional):

All reviewer comments have been adequately addressed in the revised manuscript in this editor's opinion.

Reviewers' comments:

Reviewer's Responses to Questions

**Comments to the Author**

1. If the authors have adequately addressed your comments raised in a previous round of review and you feel that this manuscript is now acceptable for publication, you may indicate that here to bypass the “Comments to the Author” section, enter your conflict of interest statement in the “Confidential to Editor” section, and submit your "Accept" recommendation.

Reviewer #1: All comments have been addressed

Reviewer #2: (No Response)

2. Is the manuscript technically sound, and do the data support the conclusions?

Reviewer #1: (No Response)

Reviewer #2: No

3. Has the statistical analysis been performed appropriately and rigorously? 

Reviewer #1: (No Response)

Reviewer #2: No

4. Have the authors made all data underlying the findings in their manuscript fully available?

Reviewer #1: Yes

Reviewer #2: Yes

5. Is the manuscript presented in an intelligible fashion and written in standard English?

Reviewer #1: Yes

Reviewer #2: No

6. Review Comments to the Author

Reviewer #1: (No Response)

Reviewer #2: General Comments:

The authors evaluated the performance of the STANDARD Q COVID-19 Ag Test (a rapid antigen test) against qRT-PCR (the gold-standard laboratory diagnostic test for SARS-CoV2 as recommended by WHO) among symptomatic patients suspected of Covid-19, who presented to primary and tertiary health care facilities in Novi Sad, Serbia. The authors observed a strong agreement between performance of the rapid antigen test and qRT-PCR.

The authors acknowledged the importance, timing and novelty of such a validation in their setting and globally. Significant changes were made to the manuscript from previous review. However, this manuscript could benefit from further review for scientific-writing errors. Examples include, but not limited to:

1. Repetition of words with resulting long text – see Lines 85,117, 143, 284 etc.

2. Slightly confusing presentation of results and discussion section – See Lines 212 -214, 217 – 220, 238-241, 257 (which of the two statistical tests was used, or both were used and what is the possible explanation for that), 314 – 318 etc.

The study could also benefit from a rigid study design for ease of reproducibility, reduction of bias and generalizability. It would be helpful for the authors to briefly explain how the sample size of 120 was determined. Was this a pre-specified/ determined sample size? The statement in Line 109 should be clarified. Asymptomatic patients were excluded from which “analysis” or were they pre-specified to be excluded from the study as stated in Line 352. Based on what should close contacts (could be asymptomatic or symptomatic) of index cases in families and “collectives” as stated Lines 340- 343 be tested using the RDT, when the study population was only symptomatic patients? It would have been helpful for the authors to also cite reference(s), criteria or case definitions used for the determination of the study population (i.e., patients with mild and moderate signs and symptoms).

7. PLOS authors have the option to publish the peer review history of their article (what does this mean?). If published, this will include your full peer review and any attached files.

Reviewer #1: No

Reviewer #2: **Yes: **Robert Samuels

---

## [Editor Report · Acceptance letter]

12 Feb 2021

PONE-D-20-34883R1 

Validation of the STANDARD Q COVID-19 Antigen Test in Vojvodina, Serbia 

Dear Dr. Ristić:

I'm pleased to inform you that your manuscript has been deemed suitable for publication in PLOS ONE. Congratulations! Your manuscript is now with our production department. 

Kind regards, 

on behalf of

Dr, John Schieffelin 

Academic Editor

PLOS ONE